# Congenital Rift Valley fever in Sprague Dawley rats is associated with diffuse infection and pathology of the placenta

**Cynthia M. McMillen**[1,2], **Devin A. Boyles**[1], **Stefan G. Kostadinov**[3], **Ryan M. Hoehl**[1], **Madeline M. Schwarz**[1,2], **Joseph R. Albe**[1,2], **Matthew J. Demers**[1], **Amy L. Hartman** [1,2]*

**1** Center for Vaccine Research, University of Pittsburgh, Pittsburgh, Pennsylvania, United States of America, **2** Department of Infectious Diseases and Microbiology, University of Pittsburgh School of Public Health, Pittsburgh, Pennsylvania, United States of America, **3** Department of Pathology, Magee Women's Hospital of the University of Pittsburgh Medical Center, Pittsburgh, Pennsylvania, United States of America

* hartman2@pitt.edu

## Abstract

Rift Valley fever (RVF) is a disease of animals and humans associated with abortions in ruminants and late-gestation miscarriages in women. Here, we use a rat model of congenital RVF to identify tropisms, pathologies, and immune responses in the placenta during vertical transmission. Infection of late-gestation pregnant rats resulted in vertical transmission to the placenta and widespread infection throughout the decidua, basal zone, and labyrinth zone. Some pups from infected dams appeared normal while others had gross signs of teratogenicity including death. Histopathological lesions were detected in placenta from pups regardless of teratogenicity, while teratogenic pups had widespread hemorrhage throughout multiple placenta layers. Teratogenic events were associated with significant increases in placental pro-inflammatory cytokines, type I interferons, and chemokines. RVFV displays a high degree of tropism for all placental tissue layers and the degree of hemorrhage and inflammatory mediator production is highest in placenta from pups with adverse outcomes. Given the potential for RVFV to emerge in new locations and the recent evidence of emerging viruses, like Zika and SARS-CoV-2, to undergo vertical transmission, this study provides essential understanding regarding the mechanisms by which RVFV crosses the placenta barrier.

## Author summary

Rift Valley fever virus (RVFV) infections cause human health and economical burdens given its ability to induce high rates of abortions in ruminants and possible contributions towards late-term miscarriages in women. In this study, we have identified important structures in the placenta targeted by this emerging bunyavirus. Inflammation was associated with more severe fetal outcomes such as death and fetal deformities. The striking similarities between the pathologies of the placenta in the rat model of congenital RVF and those observed in naturally infected ruminants highlight the utility of this rodent model.

**Data Availability Statement:** All relevant data are within the manuscript and its Supporting Information files.

**Funding:** This work was supported by the National Institutes of Health (R01AI150792 to A.L.H.; T32 AI060525 to C.M.M). The funders had no role in study design, data collection and analysis, decision to publish, or preparation of the manuscript.

**Competing interests:** The authors have declared that no competing interests exist.

These findings may be further translated towards understanding the mechanisms involved in vertical transmission of RVFV in humans.

## Introduction

Rift Valley fever (RVF) is a zoonotic disease of ruminants and humans caused by infection with Rift Valley fever virus (RVFV). RVFV is a negative-stranded, tri-segmented arbovirus of the *Bunyavirales* order (family: *Phenuiviridae*, genus: Phlebovirus) that is endemic to regions in Africa and the Arabian Peninsula. Outbreaks peak during the rainy seasons due to increases in competent mosquito vector populations (*Aedes* & *Culex* spp.) [1,2]. Rainy seasons can wreak havoc on agricultural industries due to heightened transmission of RVFV to cows, sheep, and goats leading to high rates of fetal abortions (10–40%, 10–60%, and 25–90%, respectively) and neonatal death. Susceptibility to death after RVFV infection occurs in an age-dependent manner; neonates are significantly more susceptible than adult animals [3]. Newborn and older lamb mortality can reach approximately 90% and 30%, respectively [4]. Cause of death in adult, juvenile, and fetal livestock is due to severe hepatic necrosis resulting from high levels of viral replication within the liver [5,6]. Live-attenuated vaccines are used in endemic regions to reduce the burden of disease in ruminants and the subsequent spread to people [7,8].

Still-borne lambs and calves born to mothers infected with RVFV show signs of arthrogryposis, muscle atrophy, hydranencephaly, and other nervous and musculoskeletal defects [5,6,9]. Pathologies such as raised caruncles and mineralized endometrium have also been documented in the uterus of naturally [10] and experimentally [11] infected sheep. Most reports of congenital RVF have documented gross physical abnormalities; few studies conducted detailed histological analysis of placenta in ruminants [12–14]. Vaccination of pregnant livestock with live-attenuated strains of RVFV can result in fetal teratogenicity from the vaccine strain itself [15]; therefore, vaccinations are restricted to non-pregnant animals. A major effort is underway to develop improved agricultural and human RVFV vaccines, including human clinical trials for the live-attenuated MP-12 vaccine. For these reasons it is important to understand the mechanisms of congenital disease in both livestock and humans.

Like ruminants, humans can become infected with RVFV through mosquito bite. Alternatively, farmers, butchers, and veterinarians are more likely to contract RVF from contact with contaminated tissues and virus-containing aerosols during outbreaks in livestock [16]. Human RVF is generally acute, resulting in intense fever, malaise, dizziness, and headache. Severe cases (<1%) can result in hepatic disease, hemorrhage, or late onset encephalitis, all of which can be fatal [17].

Minimal epidemiological data is available concerning the risk of vertical transmission and teratogenicity in pregnant women. Vertical transmission during the third trimester of human pregnancy was documented in two RVFV-infected women with classical signs of RVF (headache, fever, dizziness) [18,19]. These reports were limited to symptomology, serology, and gross observations of hepatic disease in both the infant and mother. A larger-scale epidemiological study performed in 2011 was the first to find a significant association between acute RVFV infection in women during pregnancy and late-term spontaneous abortion or stillbirths (odds ratio [OR], 7.4) [20]. Infection of second trimester [21] and full-term [12] human placental explants confirmed susceptibility of human placental villi to RVFV infection. These recent clinical findings highlight the importance of understanding adverse outcomes resulting from RVFV infection during pregnancy.

To provide a reliable animal model to study vertical transmission, we developed a rat model of congenital RVF using Sprague Dawley (SD) rats infected with RVFV during late gestation (embryonic day 14; E14) [21]. Vertical transmission of RVFV was observed in the placentas and fetuses of immunocompetent pregnant rats. While some infected dams succumbed to lethal RVF disease, other RVFV-infected dams without observable signs of illness delivered still-borne pups with gross physical abnormalities (i.e., stunted development, necrosis, and fetal hydrops). RVFV infection led to higher pup mortality (2.5x) in surviving dams, and viral RNA could be detected in the placenta as early as 2 days post infection (dpi). This model recapitulates many outcomes observed during natural ruminant infection [5,6] and provides an important tool to study congenital RVF and screen vaccines for teratogenic effects.

Despite abortion storms being a distinguishing outcome during RVF outbreaks, little is known about the mechanism of vertical transmission and the host immune response to infection during pregnancy. Here, we identify cellular targets of RVFV at the maternal-fetal interface and evaluate the innate immune responses to infection using the SD rat model. Furthermore, we compared the uterus and placentas from RVFV infected dams that either delivered pups with normal physical appearance to those that delivered still-borne pups with apparent physical abnormalities to identify pathologies associated with more severe outcomes and teratogenicity.

## Results

### Study design

Infection of late-gestation SD rats with virulent RVFV can serve as a tractable model of congenital RVF [21]. For the current study, we used this model to identify cellular targets of RVFV within the placentas of infected dams and to determine whether teratogenicity is associated with specific pathologies and/or immune responses to infection. On embryonic day 14 (E14) of a 22-day gestational period, we subcutaneously inoculated timed-pregnant SD rats with wild-type RVFV (strain ZH501) in the hind flank (dose range 75–1.5x10$^5$ pfu) (**Fig 1A**). Inoculation on E14 recapitulates a late-gestation infection because the rat placenta fully forms at approximately E11-12 [22] and serves as a model for the late miscarriages (second or third trimester) observed in women infected with RVFV [20]. Throughout the study, RVFV-infected dams and uninfected (no infection (NI) controls) were monitored for clinical signs of disease and euthanized if criteria were met. Uninfected controls were euthanized at 5dpi (E19; n = 3) or 6dpi (E20; n = 2). For dams that survived infection, pup delivery occurred at E22 and term placentas were collected, if available. The study concluded when surviving dams were euthanized at 18 or 22dpi.

### RVFV causes fetal infection regardless of maternal mortality

Rift Valley fever in adult non-pregnant SD rats is generally associated with extensive infection, severe hemorrhaging, and necrosis of the liver, leading to fulminant hepatic disease and a fatal outcome in 18/32 (56.2%) of rats. In pregnant rats, we surprisingly found no association between infection dose and survival (**S1A Fig**). Regardless of inoculation dose, RVFV infection resulted in around half (57%) of the pregnant dams succumbing to disease with clinical signs (hypothermia, ataxia, labored breathing) requiring euthanasia between 2-6dpi (**S1A Fig**). Fetuses from dams that were euthanized at 6dpi (E20) had gross signs of teratogenicity, such as fetal hydrops, growth restriction and hemorrhage. For dams that were euthanized from 2-5dpi (prior to 6 dpi), when the pups were E16-E19, physical abnormalities were not observed. It is unclear whether gross physical changes do not appear until later in gestation, only after longer exposures to the virus, or whether gross signs of teratogenicity are simply unable to be

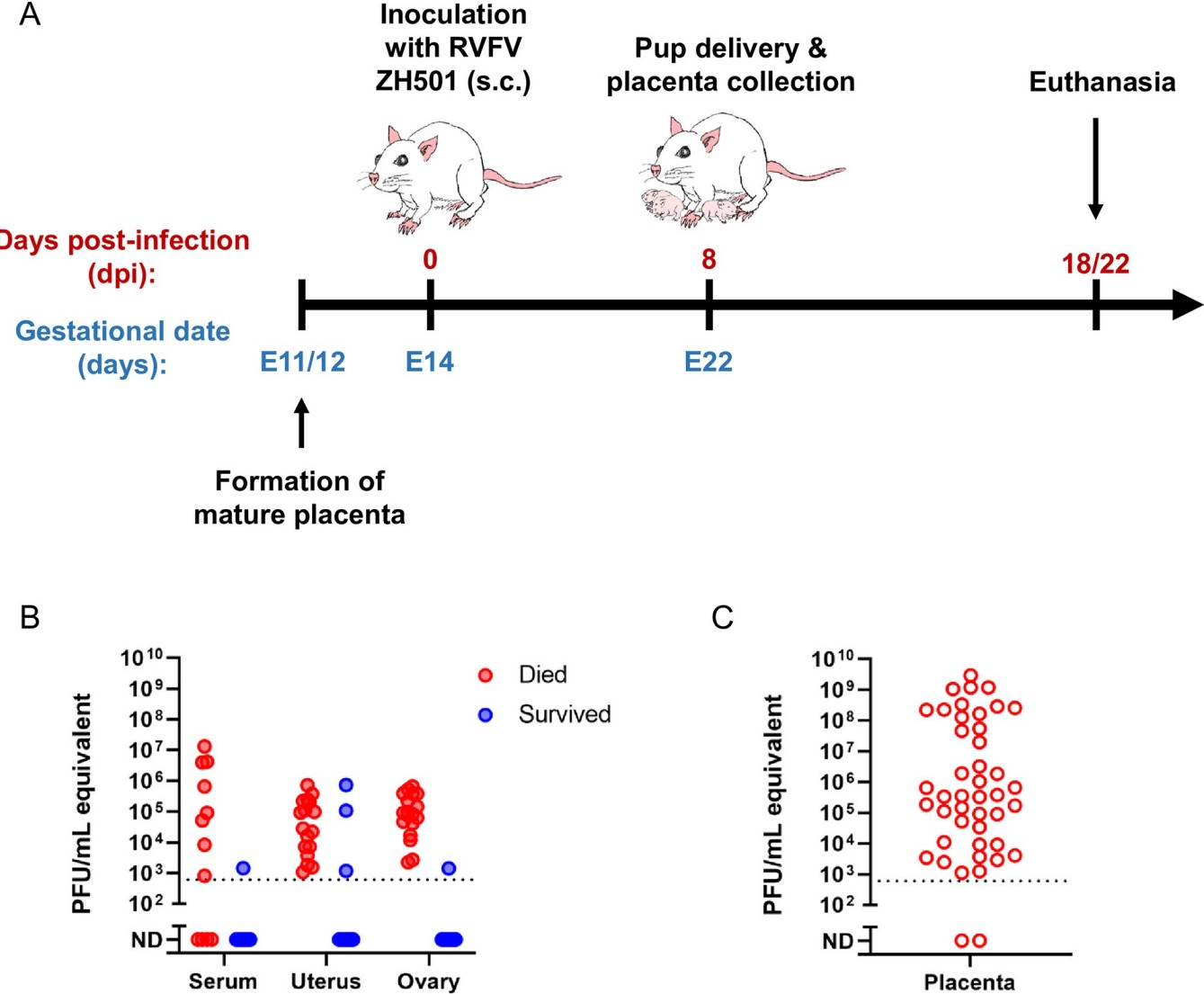

**Fig 1. Model of congenital Rift Valley fever resulting in vertical transmission in pregnant Sprague Dawley rats. (A)** Sprague Dawley (SD) rats (n = 32) were inoculated subcutaneously (s.c.) with RVFV ZH501 at embryonic day 14 (E14). Eight days later (E22), the rats delivered their pups and placentas were collected for analyses. Dams who survived infection were euthanized at 18- or 22-days post infection (dpi); serum and organs were collected for analyses. Rats meeting euthanasia criteria were immediately euthanized; dam organs, serum, and placentas were collected for analyses. **(B)** Viral RNA levels by RT-PCR (pfu/mL equivalent) of serum (n = 12 & 11, respectively), uterus (n = 17 and 15, respectively), and ovary (n = 18 and 15, respectively) of infected dams who met euthanasia criteria (died; 2-6dpi) or survived (18-22dpi). **(C)** Viral RNA levels in placenta samples collected between E20-E22 (n = 44 placentas from n = 17 dams).

distinguished by the naked eye because the fetuses are too small and/or underdeveloped prior to E20.

Dams that succumbed to infection between 2-6dpi developed viremia and high levels of viral RNA within the uterus and ovaries (**Fig 1B**). Exceptionally high levels of viral RNA (avg. $1.8 \times 10^8$ pfu/mL equivalent) were detected in placentas collected from dams at E20 or E22 (**Fig 1C**).

Of the approximately 43% of dams that survived infection after receiving doses spanning between 75 and $1.5 \times 10^5$ pfu, none displayed clinical signs of illness. Despite this, 21% of the surviving dams without clinical signs of disease delivered deceased pups. In dams that survived

to the end of the study (18–22 dpi), some residual viral RNA was still detectable in the uterus and ovaries in a few (3 of 15 and 1 of 15, respectively) of the rats (**Fig 1B**). The uterus and placenta of infected dams displayed considerable signs of pathology and RVFV viral staining, based on H&E and RNA *in situ* hybridization (ISH) (**S1B and S1C Fig**). The ovaries did not show signs of pathology nor evidence of viral infection via ISH despite detection of vRNA by RT-PCR.

## RVFV targets multiple structures within the rat placenta

We performed ISH to identify cell types and structures within the placenta that are infected by RVFV upon vertical transmission (**Fig 2A**). The rat placenta is a fetal-derived organ that is divided into two main structures, the labyrinth zone and basal zone. The labyrinth and basal zones consist of specialized trophoblast cells. The labyrinth zone contains intertwined maternal and fetal blood that is separated by cyto- and syncytiotrophoblasts. The labyrinth zone supplies the primary means of nutrient and oxygen exchange between the mother and fetus. The basal zone secretes important pregnancy regulating hormones and consists of spongiotrophoblasts, trophoblast giant cells and glycogen giant cells. At the maternal-fetal interface, the maternal-derived decidua is a specialized structure of the uterus and site of embryo implantation. The decidua, basal zone, and labyrinth zone are the main layers of the placenta and the primary maternal-fetal interface. These layers are the focus of our studies below. The yolk sac is a gestational membrane that provides supplementary means of nutrient and oxygen to the fetus.

Broadly speaking, RVFV RNA was detected in yolk sac epithelial cells (**Fig 2B, panel i**) in addition to stromal cells of the decidua and trophoblast giant cells of the basal zone (**Fig 2B, panel ii**). RVFV vRNA was also detected in the cytotrophoblasts surrounding the maternal vasculature (**Fig 2B, panel iii**). Trophoblast infection within the labyrinth zone appears to be diffuse, targeting both cyto- and syncytiotrophoblasts (**Fig 2B, panel iv**). Immunofluorescent staining for RVFV glycoprotein Gn resembles the staining observed via ISH analyses (**S2B Fig**). Placentas from uninfected dams were negative for RVFV vRNA (**S2A Fig**) and Gn (**S2C Fig**). Thus, we were able to detect RVFV infection in the 3 main layers of the rat placenta (decidua, basal zone, and labyrinth zone).

## Temporal tropism and histopathological analysis of RVFV vertical transmission in the placenta

Placentas collected from E16-E20 were evaluated to identify the placenta structures targeted by RVFV infection. RNA ISH analyses showed RVFV vRNA in the decidua, basal zone, and labyrinth zone as early as 2dpi (E16; **Fig 3**). The most substantial vRNA staining in all 3 regions was found at 6 dpi (E20).

To identify placenta pathologies associated with RVFV vertical transmission in SD rats, we analyzed H&E-stained placenta tissue sections from infected and uninfected dams at different stages of infection (**Fig 4**). Overall, among all placenta examined, RVFV infection was associated with significant levels of hemorrhage and necrosis, both of which were detected as early as 3 dpi (E17), compared to placenta from uninfected control animals. Hemorrhage severity increased over time (**Fig 4A and 4B**). When comparing the 3 main placental sections (decidua, basal zone, and labyrinth zone), hemorrhage was detected in all sections, with the labyrinth zone having the most abundant signs of congestion (**Fig 4C**). Approximately 90% of the RVFV infected placenta sections collected from E16-E20 contained hemorrhage in the labyrinth zone (**Fig 4C**). Cellular necrosis, on the other hand, occurred most frequently in the maternal decidua (98%), followed by the basal and labyrinth zones (47% and 10%, respectively; **Fig 4C**).

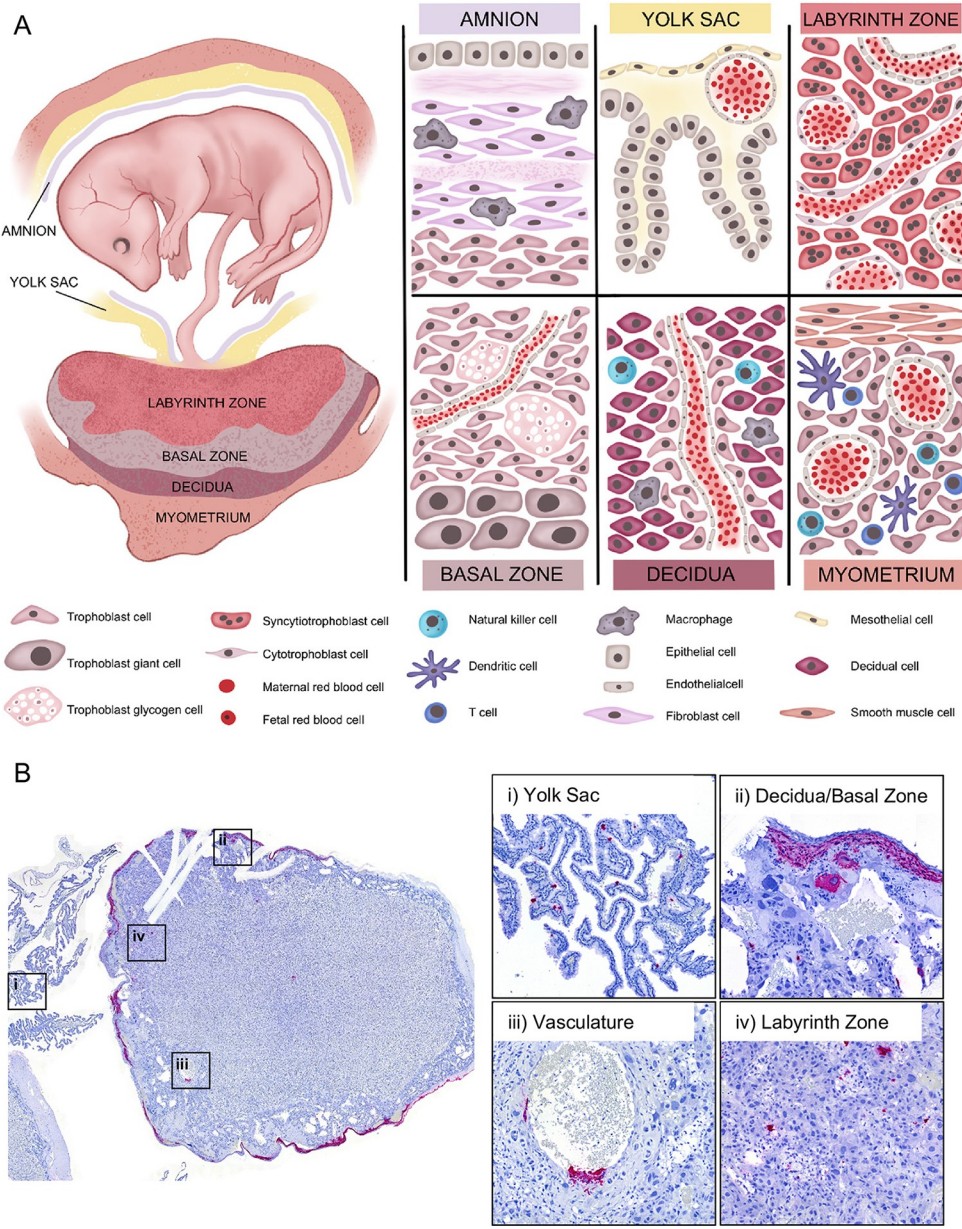

**Fig 2. RVFV infects multiple placenta structures. (A)** Maternal-fetal interface of the rat. During later stages of gestation (E14-E22) the rat fetus is surrounded by gestational membranes, the **amnion** and **yolk sac**, and connected directly to the placenta via the umbilical cord. The **amnion**, consisting of epithelium, fibroblasts, and resident macrophages, is one of many protective barriers of the fetus. The **yolk sac**, which is made of meso- and epithelial cells and capillaries, provides supplementary means of nutrient and oxygen exchange [22]. The **labyrinth zone** is the section of the fetal derived placenta in closest proximity to the fetus. This section of the placenta consists of maternal and fetal blood supplies that are separated by three distinct layers (hemotrichorial) of trophoblasts: one layer of cytotrophoblasts surrounding the maternal blood supply, and two layers of multinucleated syncytiotrophoblasts that line the fetal blood supply. The intertwined structure of the labyrinth zone provides the primary means of nutrient, oxygen, and waste exchange between the mother and fetus. The **basal zone** (also called junctional zone) is also a fetal derived tissue that helps to maintain pregnancy through production of steroids and hormones [23,24]. This section of placenta contains spongiotrophoblasts, trophoblast giant cells, and trophoblast glycogen cells. The **decidua** is at the maternal-fetal interface and is the site of embryo implantation on the uterus. The decidua is made up of decidualized uterine epithelial cells and trophoblasts. There are two subsets of trophoblasts in the decidua, interstitial trophoblasts, found throughout the decidua, and endovascular trophoblasts, lining the decidual vasculature. The **myometrium** is a distinct layer of smooth muscle within the uterus. The decidua and myometrium contain resident and infiltrating immune cells such as natural killer cells, dendritic cells, macrophages, and T cells. Under a portion of the myometrium is the metrial gland which is made up of trophoblasts and many immune cells, including natural killer cells. **(B)** Left panel: cross-section of a placenta (E20) from an infected rat with ISH staining for viral RNA (magenta). The following structures were imaged for the right panels: (i) yolk sac, (ii) decidua and basal zone, (iii) maternal vasculature, and (iv) labyrinth zone. Tissues were hematoxylin counterstained. 20x images.

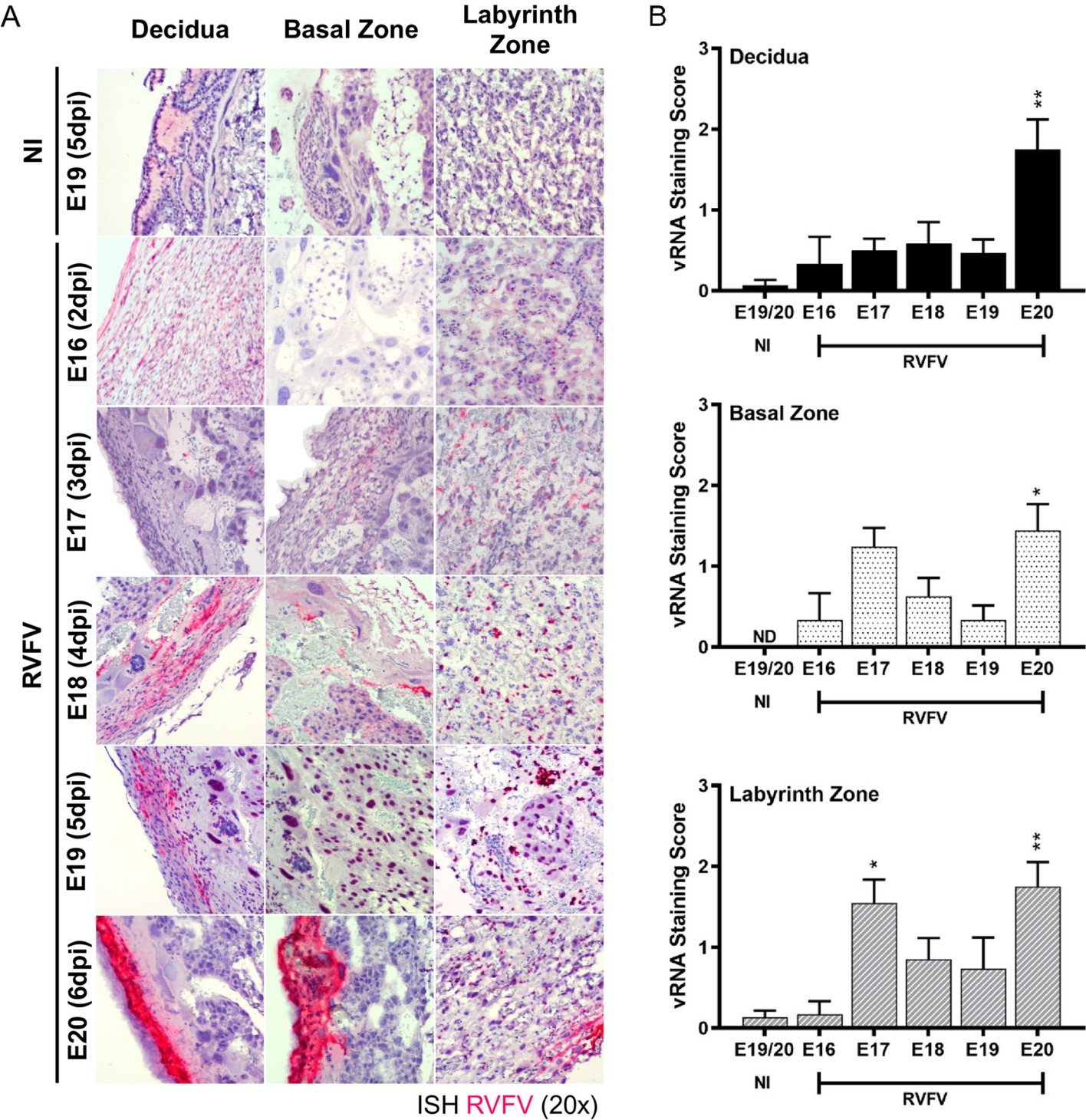

**Fig 3. RVFV infection is detected in the placenta of dams that succumb to infection as early as E16 (2dpi). (A)** Visualization of RVFV vertical transmission in placentas (n = 2, 10, 8, 5, 12, respectively) collected at E16-E20 (2-6dpi) based on ISH staining for RVFV RNA (magenta). N = 5 placentas were collected for uninfected (NI) controls **(B)** Severity of ISH staining (scale: 0–3) within the decidua (top), basal zone (middle), and labyrinth zone (bottom). NI = no infection. $^*$ = $p < 0.05$, $^{**}$ = $p < 0.01$, ND = none detected. An ANOVA with multiple comparisons was performed to determine statistical significance between the uninfected cohort and infected groups at each time point.

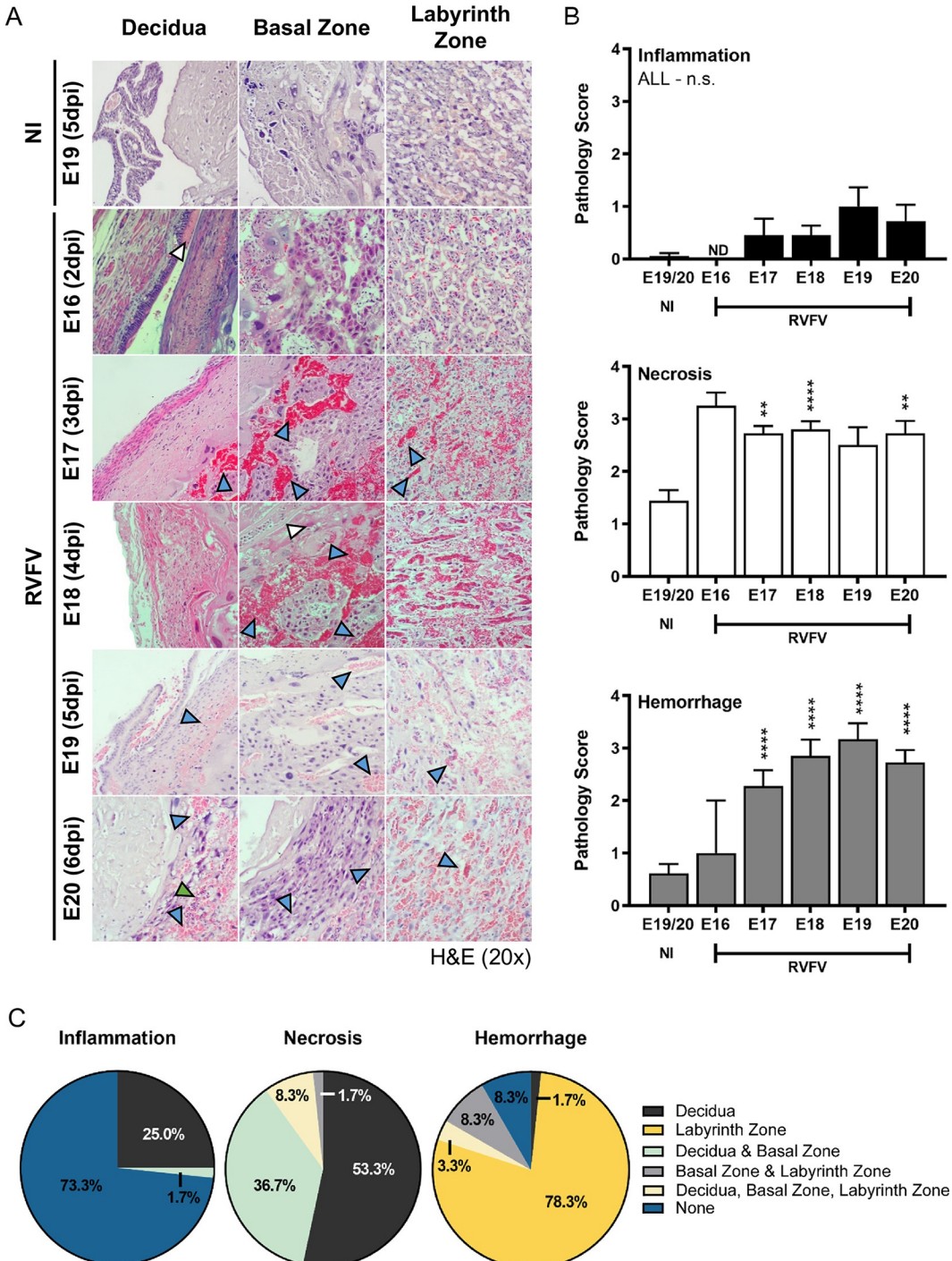

**Fig 4. RVFV-mediated pathology in the placenta of dams that succumb to infection as early as E16 (2dpi). (A)** Pathology of infected placentas (n = 2, 11, 20, 6, 11, respectively) from dams succumbing to disease from E16-E20 (2-6dpi) based on H&E staining. N = 18 placentas were collected for uninfected (NI) controls. Blue, white, and green arrow heads indicate hemorrhaging, necrosis, or leukocyte inflammation, respectively. **(B)** Pathology scores (scale: 0–4) of inflammation (top), necrosis (middle), and hemorrhaging (bottom). **(C)** Pie graph representing the percentage of placentas from all gestations (E16-22) with evidence of inflammation, necrosis, or hemorrhage in the decidua, basal zone, labyrinth zone, or multiple layers from RVFV infected dams. NI = no infection. ** = p < 0.01, **** = p < 0.0001, & n.s. = not significant. ND = none detected. In (B), an ANOVA with multiple comparisons was performed to determine statistical significance between the uninfected cohort and infected groups at each time point.

In the decidua, decidualized uterine epithelial cells showed signs of necrosis, while trophoblast giant cells were the primary necrotic cell type in the basal zone (**Fig 4A**). Overall, the placenta displayed minimal signs of inflammation, with leukocytes present primarily in the decidua (27%) (**Fig 4C**). Inflammation occurred much less frequently in the basal zone (2%) and was not observed in the labyrinth zone.

## Placentas from teratogenic pups have higher viral staining and more diffuse infection

To determine whether viral burden and certain pathologies within the placenta contribute to teratogenicity, we directly compared the placentas (E19-E22) from infected dams that delivered pups with normal physical appearance (herein "normal") and those that delivered pups with visible signs of teratogenicity (i.e. fetal hydrops, smaller size, hemorrhage; herein "teratogenic"). Teratogenicity was associated with more severe (score >2) and diffuse vRNA staining of the myometrium/decidua and basal zone, compared to placentas derived from pups with normal appearance (**Fig 5A and 5B**). Viral staining within the labyrinth zone was not significantly different between placentas from normal and teratogenic pups (**Fig 5B**).

Both normal and teratogenic cohorts had significant levels of necrosis and hemorrhage compared to placentas from uninfected controls (**Fig 6A and 6B**). In contrast, inflammation was rarely detected in any placentas collected from E18-E22 regardless of the presence of teratogenicity. Surprisingly, there were no differences in inflammation, necrosis, nor hemorrhage pathology scores between teratogenic and normal pups (**Fig 6B**). However, a major difference between teratogenic and normal placentas was that hemorrhage was more diffuse across multiple placental layers from teratogenic pups (57.2%), whereas hemorrhage was limited to the labyrinth zone in placentas from pups with normal appearance (7.5%; **Fig 6C**). Conversely, necrosis was found in multiple placental layers in placentas from normal pups (47.5%) while necrosis was more limited to the decidua in placenta from teratogenic pups (14.3%; **Fig 6C**).

## Elevated cytokine and chemokine expression is associated with teratogenic pregnancies

To better understand the inflammatory responses during RVFV infection in pregnant rats, cytokine and chemokine gene and protein expression were evaluated in the placentas of infected and uninfected dams collected at E20-E22. Infected placentas were further divided to identify differences between placentas derived from pups with normal physical appearance and those from pups with observable signs of teratogenicity. Overall, RVFV infection was associated with increased levels of IFNλ3 mRNA in infected teratogenic placentas compared to uninfected controls (**Fig 7A**). Differences in IFNλ3 mRNA expression was not found between normal and teratogenic placentas from infected dams. IFNβ mRNA was slightly elevated in placentas from teratogenic pups compared to placentas from normal pups and uninfected controls (**Fig 7A**).

When further comparing the protein levels of inflammatory mediators in infected placentas from either normal or teratogenic pups, IL1α, MCP-1/CCL2, MIP-1 α/CCL3, RANTES/CCL5, and GRO/KC/CXCL1 were significantly elevated in the placentas from teratogenic pups but not in placentas from pups with normal appearance, compared to uninfected controls (**Fig 7B**). Furthermore, the same cytokines and chemokines were significantly elevated in teratogenic placentas compared to normal placentas. For example, RANTES/CCL5 was approximately 10x higher in teratogenic placentas than infected placentas from pups with normal appearance. IL-18 protein expression was also elevated (p = 0.145 & 0.113, respectively) in

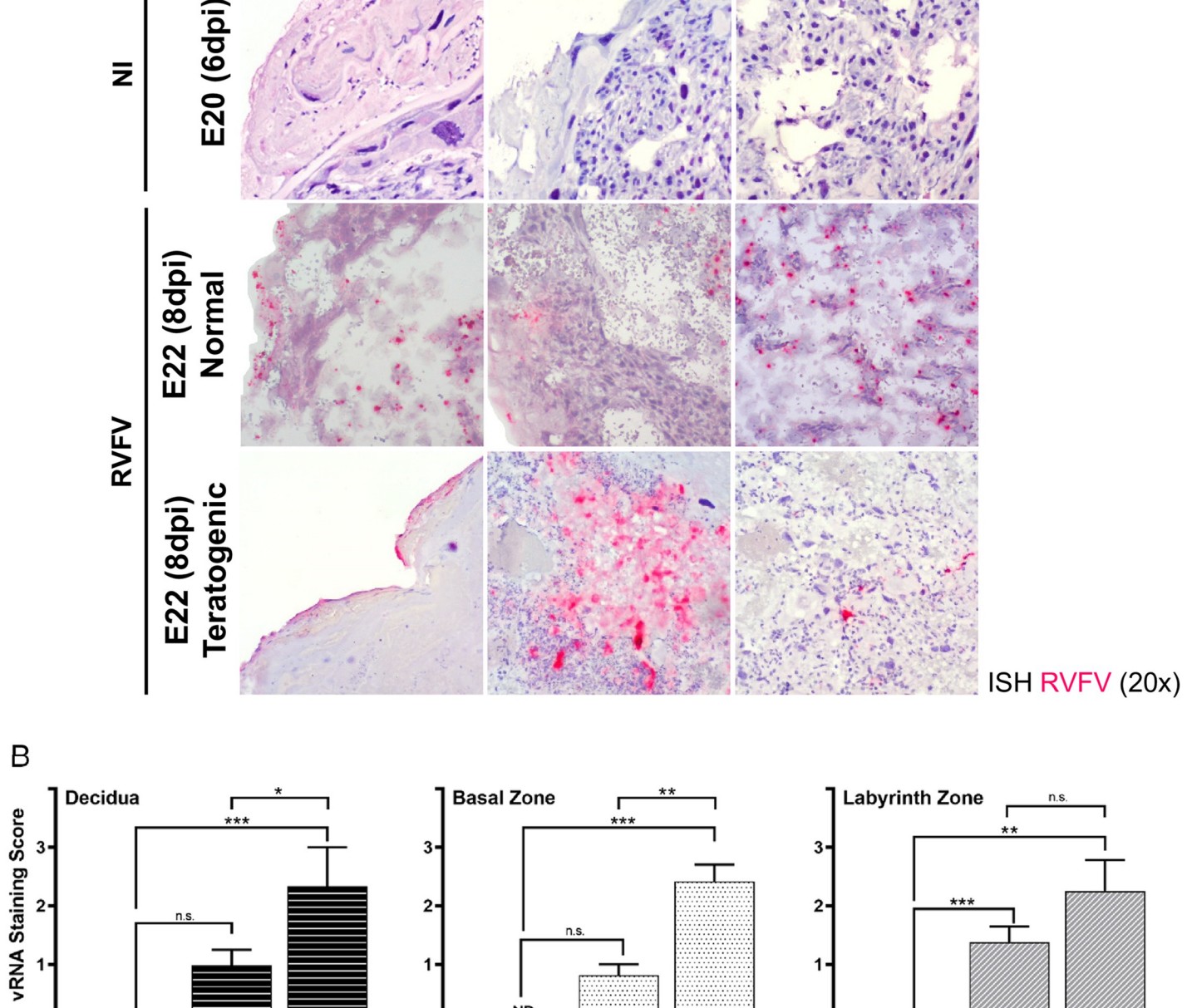

**Fig 5. Teratogenicity is associated with widespread infection of multiple placenta layers. (A)** ISH (magenta) stained placentas (E20 & E22) from uninfected (NI) and infected dams. Infected placentas are segregated into those from pups with normal appearance (herein "normal") or those from pups with gross signs of teratogenicity (herein "teratogenic"). Panels are separated based on placenta regions: decidua, basal zone, and labyrinth zone. Tissues were hematoxylin counterstained. 20x images. **(B)** Severity of ISH staining (scale: 0–3) in normal and teratogenic pups based on individual regions of placentas collected between E19-22. * = p < 0.05, ** = p < 0.01, *** = p < 0.001, & n.s. = not significant. ND = none detected. An ANOVA with multiple comparisons was performed to determine statistical significance between groups. NI (n = 6), normal (n = 15), teratogenic (n = 4).

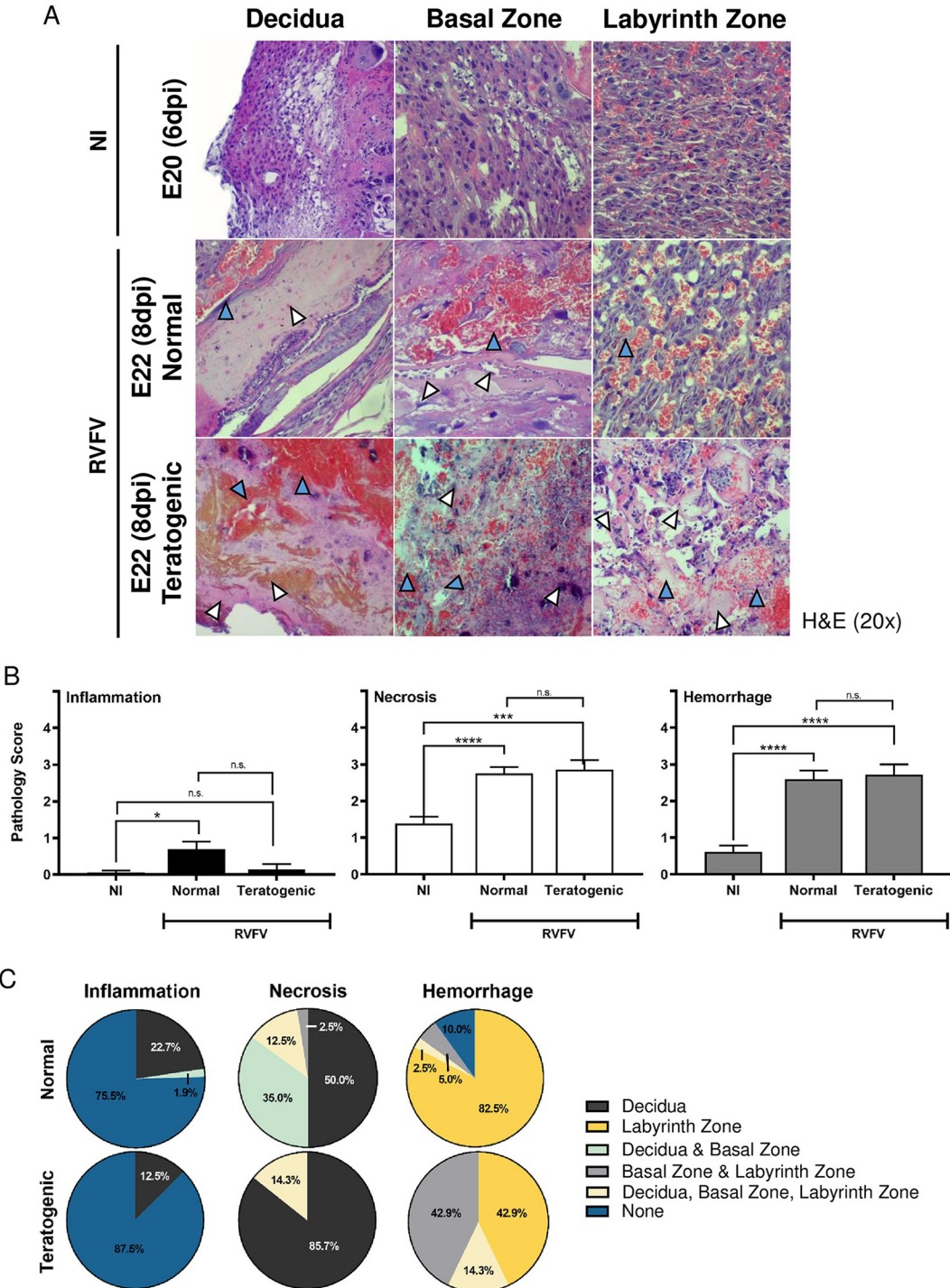

**Fig 6. Widespread hemorrhaging in the placenta is associated with teratogenic pups. (A)** H&E placentas from uninfected (NI) and infected dams that delivered normal pups or pups with gross signs of teratogenicity. Panels are separated based on placenta regions: decidua, basal zone, and labyrinth zone. Blue and white arrow heads indicate hemorrhaging or necrosis, respectively. 20x images. **(B)** Pathology scores (scale: 0–4) identifying inflammation, necrosis and hemorrhage severity of H&E stained placentas (E18-E22). **(C)** Pie graph demonstrating the percentage of placentas displaying inflammation, necrosis, or hemorrhage within the decidua, basal zone, labyrinth zone, or multiple layers. Placentas were grouped into normal or teratogenic based on gross observation. NI, no infection. $^{***}$ = p < 0.001, $^{****}$ = p < 0.0001, & n.s. = not significant. ND = none detected. An ANOVA with multiple comparisons was performed to determine statistical significance between groups. NI (n = 18), normal (n = 18), teratogenic (n = 7).

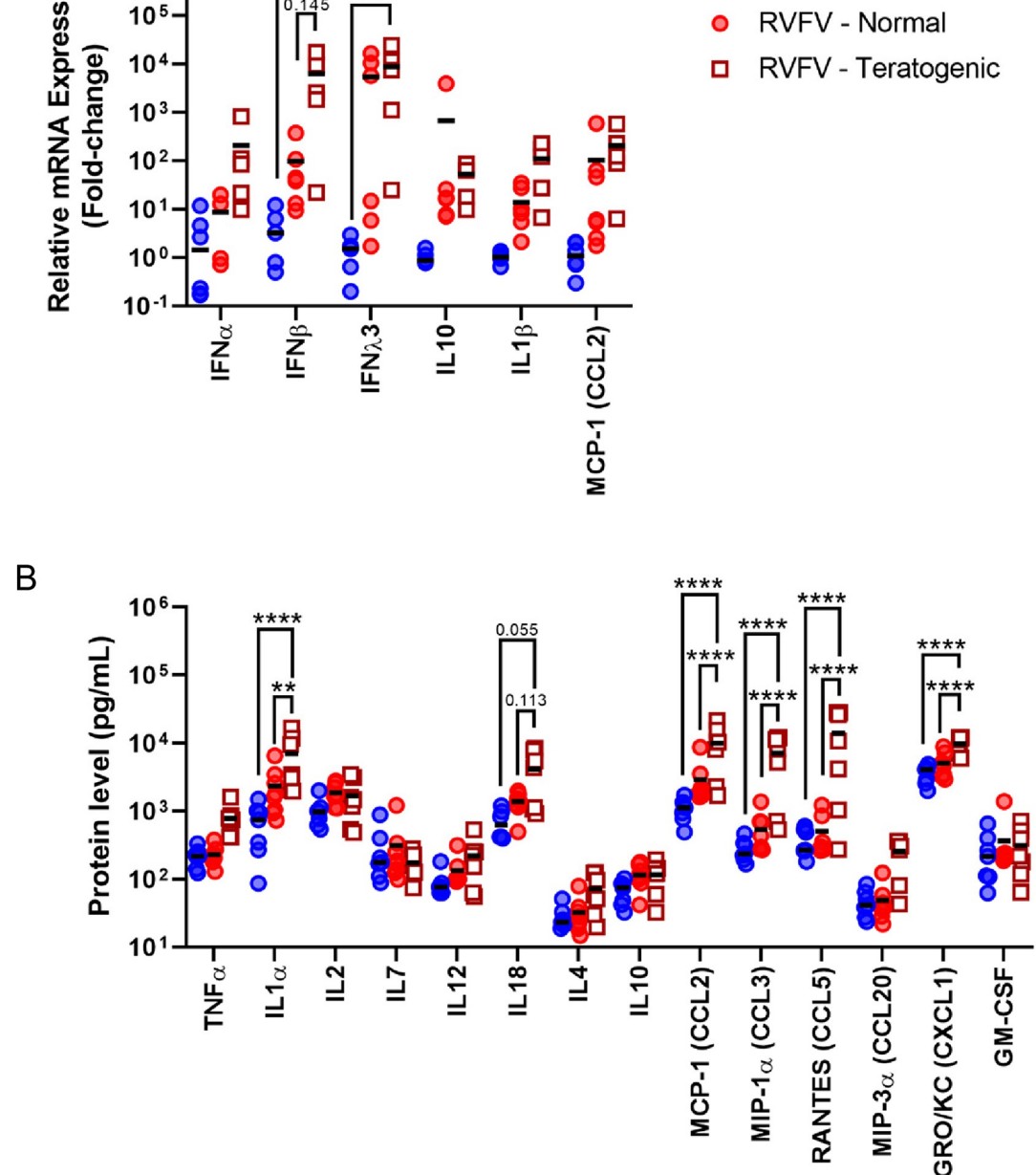

**Fig 7. RVFV infection leads to increased Th1 cytokine responses and chemokine expression within the placenta at E20-22.**
**(A)** Cytokine mRNA; uninfected (n = 5–6), RVFV–normal (n = 4–6), RVFV–teratogenic (n = 5–6) or **(B)** protein expression within placentas from uninfected dams and RVFV infected dams with normal or teratogenic pups. Mean is denoted by black horizontal line. uninfected (n = 13-6-7), RVFV–normal (n = 7–8), RVFV–teratogenic (n = 7) ** = p < 0.01, **** = p < 0.0001. An ANOVA with multiple comparisons was performed to determine statistical significance between the groups.

teratogenic placentas compared to infected but normal pup placentas (**Fig 7B**). Despite high levels of chemokines found within the placenta, we saw minimal to no inflammation in histology sections (**Fig 6**).

### Elevated cellular inflammation of the uterus occurs in RVFV infected pregnant SD rats

Besides the placenta, TORCH (toxoplasmosis, other, rubella, cytomegalovirus, and herpes virus) pathogens can also cause detrimental effects to the maternal uterus. Here, we examined uterine tissue and found that the uterus of dams that succumbed to infection between E16-20 had mild-to-moderate levels of inflammation (p<0.001) compared to uninfected controls (**Fig 8A**), whereas similar levels of hemorrhage was observed between the groups. Necrosis was not

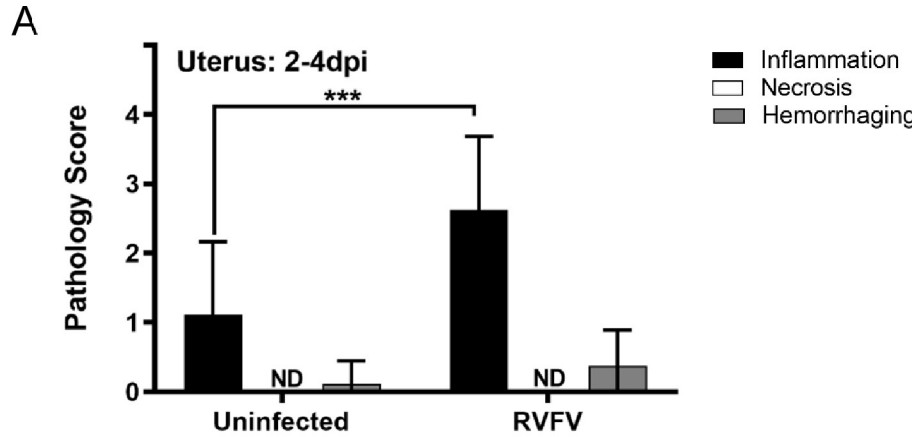

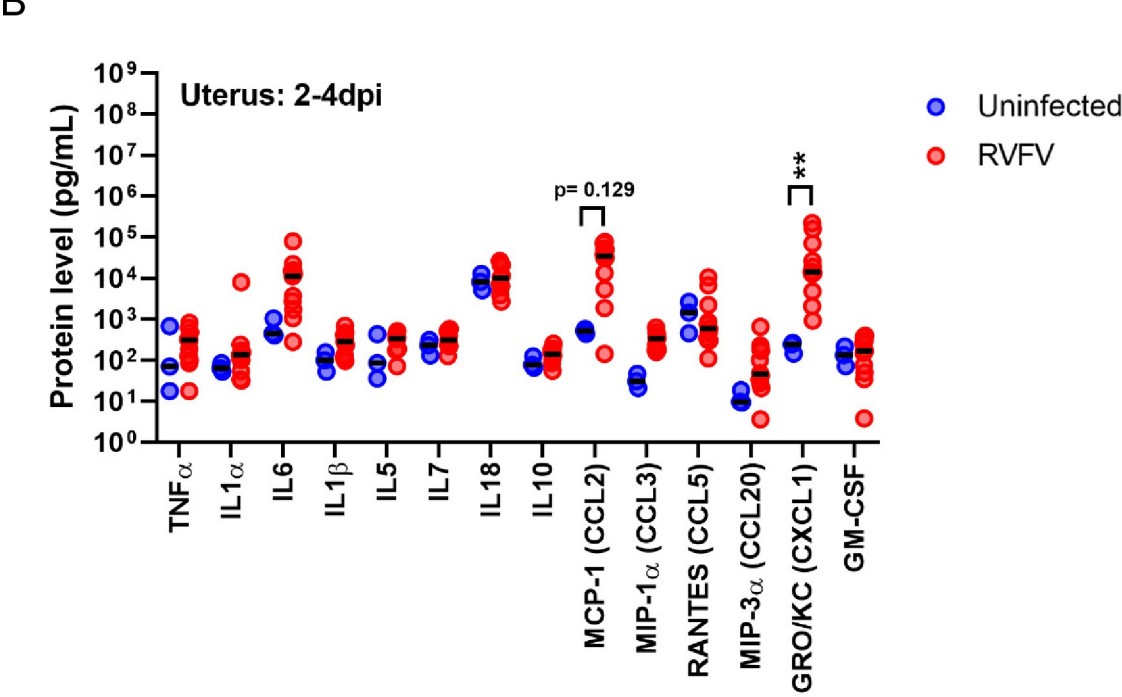

**Fig 8. The uterus from RVFV infected dams contains inflammation and elevated chemokines. (A)** Pathology scores (scale: 0–4) identifying inflammation, necrosis and hemorrhage severity of H&E stained uterus samples from dams that succumbed to infection between 2–4 dpi. Uninfected (n = 9–10), RVFV (n = 8). **(B)** Protein levels (pg/mL) of cytokines, chemokines, and growth factors within the uterus of infected and uninfected dams that succumbed to infection between 2-4dpi. Uninfected (n = 3), RVFV (n = 9–12). ** = p < 0.01, *** = p < 0.001 compared to uninfected controls. ND = none detected. An ANOVA was performed to determine statistical significance between groups.

detected in the tissue sections collected from either cohort. In line with the higher levels of inflammation, GRO/KC/CXCL1 and MCP-1/CCL2 were found at higher levels in the uterus of RVFV infected dams compared to uninfected controls (**Fig 8B**).

Inflammation remained significantly elevated in the uterus of post-partum RVFV infected dams that were euthanized at 18-22dpi compared to uninfected controls (p < 0.05; **S3A Fig**). Uninfected and infected dams had similar levels of hemorrhage in their uterus which could be due to the natural delivery process. In post-partum dams, only RANTES/CCL5 was elevated in the uteruses from infected dams that delivered pups with normal physical appearance compared to uninfected controls (**S3B Fig**). No differences in cytokine, chemokine, and growth factor expression were noted between cohorts that delivered pups with normal physical appearance and those that delivered pups with signs of teratogenicity.

## Discussion

Our findings demonstrate that RVFV infects the majority of maternal and fetal structures and cell types within the rat placenta, highlighting the highly pan-tropic nature of this virus. This is the first study to identify the cellular targets of RVFV in a rodent model of congenital RVF. Mice infected with RVFV by intrauterine injections 3–4 days prior to term (approx. 21 days) had been utilized once prior [23]; vertical transmission occurred, but gross-physical abnormalities and still-births were not observed in this model. Histological analyses were not performed on the embryos and placentas were not collected.

Based on our results, RVFV targets similar cell types of the placenta in rodents as humans and ruminants. These findings further support the use of this model to understand the course of vertical transmission and teratogenicity of RVFV in naturally infected species. For instance, *ex vivo* infection of second trimester [21] and full-term human placentas [12] showed infection of cytotrophoblasts and syncytiotrophoblasts of chorionic villi, which correlates with infection of similar cells in the labyrinth zone (cytotrophoblasts and syncytiotrophoblasts) and basal zone (giant cells) of rats. Immortalized human trophoblast cell lines (A3 and Jar), are also permissive to RVFV infection [24]. Despite general similarities in cellular constituents, there are distinct differences in the structure of placentas in rats and humans which should be considered when using rats as a model to study human disease. From a broad perspective, humans and rodents have similar placenta structures given they both have discoid hemochorial placentas [25], however the number of cells separating the maternal and fetal blood are different between species. Humans have a hemomonochorial placenta, meaning there is only one layer of syncytiotrophoblasts separating blood, whereas rodents have a hemotrichorial placenta which have two layers of cytotrophoblasts and a single layer of syncytiotrophoblasts separating maternal and fetal blood. At the maternal-fetal interface, the human placenta forms branch like structures consisting of villous trophoblasts and invasive extravillous trophoblasts located at the end of the branches that anchor the placenta to the uterine wall [26]. This branched structure is bathed in maternal blood, providing quick nutrient exchange. Rats, on the other hand, have a labyrinthine structure consisting of intertwined maternal blood and fetal vasculature [26]. Rats also have highly invasive cells, trophoblast giant cells, that embed themselves into the uterine tissue and line the junctional zone between the placenta and uterus [27,28]. These structural differences, and others not discussed, could affect pathogenic outcomes between species and limit the use as rats as a surrogate for human or ruminant disease [29,30].

Comparable placenta cell-types are also targeted by RVFV in sheep and rats. In placentas collected from sheep infected in 2010 during a natural outbreak in South Africa, RVFV antigen staining was predominantly found in the fetal villus trophoblasts, and infection of multinucleated syncytiotrophoblasts residing in the maternal caruncle, the sheep uterus, was present [14].

Infection of endothelial cells within the vasculature of the chorioallantoic membrane and sheep uterus also occurred. Another recent study by Oymans [12] has provided additional insight into the cellular targets of RVFV. Recombinant RVFV strain 35/74 inoculated in pregnant sheep at one-third gestation (embryonic day 55; E55) or mid-gestation (E78) underwent vertical transmission to the placenta and fetus as early as 4 dpi for both cohorts. RVFV appeared to establish infection within the sheep placenta by two methods: 1) by direct infection of fetal trophoblasts in the hemophagous zone, a region of the placenta where the maternal blood is in direct contact with the fetal cells, or 2) by first infecting maternal epithelium within the villi prior to spreading to the fetal trophoblasts. Due to the diffuse labyrinthine nature of placenta infection in our rats, we were unable to deduce the route of established infection within the placenta. Cells at all points of entry (i.e., decidual cells, trophoblast giant cells of the basal zone, endothelial cells and cyto- and syncytiotrophoblasts of the labyrinth zone) are permissive to infection, thus it is likely that multiple routes of invasion into the placenta could occur in rats as well. Future studies looking at placentas collected as early as 12hpi could shed light as to which cells are first targeted by RVFV.

The ruminant placenta has a vastly different structure than human placentas [31,32]. Ruminants have cotyledonary placentas consisting of mini-placentomes spanning the fetal membrane. Their synepitheliochorial cellular structure provides additional separation between maternal and fetal blood, which is a key difference in nutrient exchange between humans, rodents, and ruminants. Ruminant and rats contain trophoblast giant cells that aid in uterine implantation [32]. Structural differences and instances of conserved cellularity within placentas should be considered when using rodents as a model for ruminant disease.

Despite the outlined structural differences, pathology within the rat placenta found here displayed similarities to what is observed in ruminants. In our studies, necrosis was seen in the proximal regions of the maternal-fetal interface, most abundantly in the decidua and to a lesser extent, the basal zone. Necrosis was elevated even at early stages of infection. In the study by Oymans [12], necrosis occurred primarily in the maternal villus epithelium of RVFV infected sheep. Minimal necrosis was observed in fetal trophoblasts and endothelial cells showed no signs of necrosis despite infection. Odendaal [14], however, saw evidence of endothelial necrosis in naturally infected sheep.

The main pathological outcomes observed in RVFV-infected rat placentas were necrosis (primarily in decidua) and hemorrhage (primarily in labyrinth zone). There was surprisingly little inflammation in any placental structure. Zika virus (ZIKV) infection of pregnant IFNAR1 knockout mice via footpad injection can result in vertical transmission involving severe necrosis of the decidua and basal zone [33,34]; however, decidual necrosis in ZIKV-infected rodents and pregnant women [35] is not associated with adverse fetal outcomes such as death or microcephaly. Vertical transmission of human immunodeficiency virus in placentas from 1st and 2nd trimester abortions is also correlated with necrosis of the decidua [36]. Chronic lesions such as decidual necrosis is one of the key pathological findings associated with placenta abruption [37,38], which is a premature detachment of the placenta from the uterus that can increase the risk of fetal death [39,40]. In addition, vertical transmission of cytomegalovirus (CMV) was 4x more likely to occur in mothers with decidua abruption [41]. Decidua necrosis and premature detachment could be an etiologic factor of teratogenicity and fetal death in the SD rat model of congenital RVF but requires more detailed study.

Severe hemorrhage within the placentas of RVFV infected dams should also be considered as a major contributor to fetal death. In pregnant SD rats, severe hemorrhaging always occurred in the labyrinth zone of infected placentas; however teratogenic placentas were more likely to experience diffuse hemorrhage, appearing within two or more layers, including the labyrinth zone. Hemorrhage also occurs in placentas from ZIKV infected mice [42], whereas

placental hemorrhage is not associated with CMV infections [43]. Extensive placental hemorrhage occurs in sheep experimentally [11,12] or naturally [14] infected with RVFV. In sheep, hemorrhages occur next to the chorioallantoic villi of the sheep placentome and the uterine wall. Significant neutrophil influx was also observed in the lamina propria of the uterus, next to the maternal villus. Hemorrhaging of the placenta can occur due to a breach in endothelial cell structural integrity by virus-induced endothelial damage or over-abundant vascular permeability. Alterations in vascular permeability might account for hemorrhage seen in our model.

The lack of inflammatory cells in infected rat placentas is puzzling because we detected elevated levels of cytokines and chemokines. Cytokine and chemokines can enhance cellular inflammation into the infected region. The lack of inflammation, perhaps, may be explained by the kinetics of cytokine and chemokine expression which proceeds the influx of immune cells. Since expression levels were only analyzed during later stages of gestation (E20-E22), it is feasible that our analyses preceded inflammation. Infection earlier in gestation may result in detectible inflammation at later stages of gestation. The lack of inflammation in the placenta may have resulted in exacerbated viral replication and virus-induced pathologies that could have otherwise been controlled by the innate and early adaptive cellular responses. Inflammation at the maternal-fetal interface however can be a double-edged sword. Inflammation of placental tissue can be a major issue because influx of immune cells can result in leaky vasculature leading to hemorrhage or entry of maternal effector cells into the immune privileged fetal tissue. In fact, increased protein levels of RANTES/CCL5 and VEGF receptor (VEGF-R) have been detected in human placentas with ZIKV infection [35] and their expression are associated with vascular permeability. RANTES/CCL5 expression was 4-5-fold higher in human placentas from babies with microencephaly, compared to ZIKV infected placentas from without microencephaly [35]. RANTES/CCL5 was significantly elevated in RVFV infected placentas and teratogenic placentas had approximately 10-fold more protein than infected placentas from pups with normal physical appearance. Looking a step further, blockage of RANTES/CCL5 responses can inhibit the spread of ZIKV in human brain microvascular endothelial cells [44]. RANTES/CCL5 might have a role in vertical transmission of RVFV to the fetal brains observed in our previous study [21]. At a protein level, we did not see an increase in VEGF expression, a regulator of vascular permeability, upon RVFV infection. VEGF receptor (VEGF-R) expression was not analyzed. Elevations in this receptor might also enhance vascular permeability and lead to extensive hemorrhage.

A generalized pro-inflammatory response was detected in RVFV-infected rat placentas, including elevations in IL1α, IL-18, type I interferon (IFNβ) and chemokines such as MCP-1/CCL2 and RANTES/CCL5. Teratogenic placentas accounted for these increases along with other chemokines associated with acute-viral infections, such as MIP-1α/CCL3 and Gro/KC/CXCL1 [45]. Inhibition of type I interferon signaling in pregnant mice infected with ZIKV has shown that interferon signaling promotes fetal demise [46]. Another study showed that the immune responses to ZIKV infection alone could cause fetal demise and adverse outcomes [47]. Recurrent miscarriages have been attributed to both low [48] and high [49] levels of IL-18 expression in women. Considering IL-18 is a pyrogen, identifying whether pyroptosis contributes to the pathologies observed upon vertical transmission of RVFV to the placenta should be evaluated. ZIKV can induce pyroptosis in JEG-3 placenta trophoblasts *in vitro* [50]. Considering inflammatory cells were present in the rat uterus and decidua, and less in the basal and labyrinth zones, the increase in pro-inflammatory mediators and chemokines could be mainly attributed to expression in the maternal tissue at the maternal-fetal interface. This could also explain why most of the cell death and necrosis was observed in the decidua of the rats in this study. CCL5 expression was primarily elevated in the decidua and Hofbauer cells of ZIKV

infected placentas [35]. Flow cytometric analyses or single-cell RNA sequencing may pinpoint which cells are responsible for these elevated cytokines and chemokines. Future studies performing serial euthanasia could provide a more complete picture of the cellular inflammation and immune response present in the uterus and placenta from infection to delivery. This study is limited by the fact that the uterus was not collected upon delivery of the pups (E20-E22) when the placenta was collected, therefore direct comparisons in cytokine expression between post-partum uterus and E20-E22 placentas is unable to be performed.

Although the uterus and placenta showed signs of infection, neither pathology nor viral staining was seen in the ovaries of infected dams that succumbed to infection, despite high levels of vRNA detected by qPCR. Considering other teratogenic arboviruses, such as ZIKV, have been shown to infect the ovaries and other regions of the female reproductive systems in mice [51] and non-human primates [52], RVFV infection of the ovaries should not be ruled out and warrant further analyses.

Overall, we have identified important structures and cells in the placenta targeted by RVFV. Increased pro-inflammatory cytokine and chemokine expression was associated with more severe fetal outcomes such as teratogenicity. We have thus far only touched the surface of our understanding of RVFV infection and associated immune responses to infection of the placenta. To fully understand the impact of RVF on miscarriage rates in pregnant women, large scale population studies evaluating the effect of RVFV infection during pregnancy at various gestations is needed. Additionally, post-partum evaluation of human placenta will provide key information needed to understand the mechanism of RVFV vertical transmission and further confirmations of the relevance of our rodent model.

# Materials and methods

## Ethics statement

All animal work described here was carried out in strict accordance with the Guide for the Care and Use of Laboratory Animals of the National Institutes of Health and the Animal Welfare Act. The protocol was approved and overseen by the University of Pittsburgh Institutional Animal Care and Use Committee. The Association for Assessment and Accreditation of Laboratory Animal Care has fully accredited the University of Pittsburgh.

## Biosafety

Work with infectious RVFV (strain ZH501) was performed at biosafety level 3 (BSL-3) in the University of Pittsburgh Regional Biocontainment Laboratory (RBL). Personnel wore powered air-purifying respirators (Versaflo TR-300, 3M) for respiratory protection. All active work was performed in class II biological safety cabinets. Animals were housed in individually ventilated microisolator cages (Allentown Inc.) All waste and surfaces were disinfected with Vesphene IIse (1:128 dilution; Steris Corporation). All tissues or samples designated for removal from BSL-3 for downstream processing were inactivated using methods described below; all inactivation methods were verified and approved by a University of Pittsburgh biosafety oversight committee. The University of Pittsburgh RBL is a registered entity with the Centers for Disease Control and Prevention and the U.S. Department of Agriculture for work with RVFV.

## Cell culture

Vero E6 cells (CRL-1568, ATCC) were grown in DMEM (Dulbecco's Modified Eagle Medium: Corning) supplemented with 10% fetal bovine serum (FBS), 1% L-glutamine and 1% (v/v)

penicillin/streptomycin (pen-strep). Cells were maintained in a humid incubator at 37°C at 5% $CO_2$.

## Virus

Virulent RVFV (strain ZH501) was derived from reverse genetics plasmids [53] provided by B. Miller (CDC, Ft. Collins, CO) and S. Nichol (CDC, Atlanta). RVFV was propagated on Vero E6 cells using standard methods. Viral titer was determined by standard viral plaque assay (VPA). Prior to animal infections, stock virus was diluted in D2 (DMEM, 2% fetal bovine serum (FBS), 1% L-glutamine, 1% pen-strep) to the preferred concentration. To account for components within the D2 media, all uninfected control cohorts were mock infected with D2 media.

## Animal studies

Six-to-eight-week-old timed-pregnant SD rats were obtained from Envigo Laboratories. A positive copulation plug verified pregnancy. Animals arrived at our facility at embryonic day 12 (E12) and housed individually in temperature-controlled rooms with a 12-hour day/12-hour night light schedule. Food (IsoPro Rodent 3000) and water were provided *ad libitum*. Animals acclimated to their new surroundings for 48 hours prior to infection. At E14 the SD rats were anesthetized by inhalation of isoflurane (IsoThesia, Henry Schein) then implanted with programmable temperature transponders (IPTT-300, Bio Medic Data Systems) subcutaneously between the shoulder blades. Following implantation, the rats were injected subcutaneously (s.c.) in the hind flank with 200uL of RVFV at the following titers: $1.5x10^5$ pfu (n = 3), $2.6x10^4$ pfu (n = 6), $1.4x10^3$ pfu (n = 11), $1.8x10^2$ pfu (n = 6) and 75 pfu (n = 7). D2 media was delivered to age- and gestation-matched animals as uninfected (no infection, NI) controls (n = 5). Weight and body temperature were documented daily starting the day of infection. The SD rats were monitored twice daily for development of clinical signs of disease. Euthanasia criteria was determined by the following parameters in adherence with IACUC guidelines. Rats were euthanized if they received a combined score of 10 or above or if they received a score of a 4 in any of the criteria outlined in Table 1.

Upon RVFV infection the most prominent clinical signs of disease were hypothermia (<34.0°C), behavioral changes, and changes in appearance. Neurological signs were rare. Uninfected controls were euthanized at 5dpi (E19; n = 3) or 6dpi (E20; n = 2). The rats delivered their pups at embryonic day 22 (E22), at which time available placentas were collected. At 18- or 22-dpi dams were euthanized. Animals that met euthanasia criteria prior to this time

**Table 1. Rat health scoring criteria.**

| Score | 0 | 1 | 2 | 3 | 4 |
|---|---|---|---|---|---|
| **Weight Loss*** | up to 5% | 6–10% | 11–15% | 16–20% | >20% |
| **Temperature*** | +0.5°C | +0.6–1.0°C | +1.1–1.5°C | >+2.1°C | <34.0°C |
| **Behavior** | normal | less peer interaction | huddled | moved only when prodded | recumbent in cage or unable to stand upright or walk |
| **Appearance** | normal | ruffled fur | hunched posture | half-closed eyes or porphyrin staining around eyes/nose/mouth | respiratory distress |
| **Neurological** | normal | loss of muscle coordination or head tilt | erratic/easily spooked/jumpy movements, large circles in cage or tremors of the head | small circles in the cage | seizures including rolling in the cage |

* based on changes from baseline

were immediately anesthetized with isoflurane and euthanized by cardiac puncture, followed by perfusion using phosphate buffered saline (PBS). During necropsy, the following tissues or body fluids were collected from each dam: brain, liver, uterus, ovary, placenta, amniotic fluid. Serum was collected from each dam prior to perfusion. Half of each tissue was either frozen immediately for downstream molecular analyses or fixed in 4% PFA for histology.

## Tissue preparation

Half of each tissue was weighed and homogenized in D2 media (v/v) using an Omni tissue homogenizer (Omni International), then stored at -80˚C until downstream analyses were performed. Tissue homogenates were used to quantitate infectious virus by viral plaque assay (VPA) as described previously [21]. For quantitation of RVFV-specific viral RNA (vRNA) by semi-quantitative real-time polymerase chain reaction (RT-PCR), 100uL of each tissue homogenate or liquid sample (blood, amniotic fluid) was inactivated in 900uL of Tri-Reagent (Invitrogen) for 10 minutes prior to removal from the BSL-3 facility as per approved inactivation protocols. Subsequent storage at -80˚C or immediate RNA isolation and RT-PCR analyses were performed in a BSL-2 setting using the parameters stated in McMillen, Arora (21). The other half of each dam tissue was submerged in 4% PFA for 24 hours for fixation and virus inactivation. Fresh 4% PFA was added prior to the removal of fixed tissues from the BSL-3 laboratory. Fixed tissues were delivered to a BSL-2 setting and stored in PBS prior to embedding in paraffin and cut onto slides, using standard methods, for hematoxylin and eosin (H&E) and *in situ* hybridization (ISH) analyses. For immunofluorescence (IF) staining, fixed tissues were cryopreserved using standard methods.

## Semi-quantitative real-time PCR for detection of cytokines

For detection of cytokines (IFNα, IFNβ, IFNλ3, IL-10, IL-1β) and chemokines (MCP-I /CCL2), total RNA was first converted to cDNA using the M-MLV reverse transcriptase (Invitrogen) following the manufacturer's protocol, including the use of random primers (Invitrogen) and RNAseOUT recombinant ribonuclease inhibitor (Invitrogen). Next, semi-quantitative RT-PCR was performed using Taqman Multiplex Master Mix (2x; Applied Biosystems) and Taqman Gene Expression Assay kits (Invitrogen; **S1 Table**) following the manufacturer's instructions. Endogenous controls for normalization comprised of β-actin, while corresponding uninfected, gestation-matched tissues served as reference tissue. The thermal cycling parameters included a hold step at 95˚C for 20 seconds, then a cycling PCR amplification step including a 95˚C hold for 1 second, a 60˚C hold for 20 seconds that was repeated 40x.

## Protein quantification

Cytokines, chemokines, and growth factor protein expression was quantified using the Bio-Plex Pro Rat Cytokine 23-plex assay (Bio-Rad Laboratories, Inc). Analytes included in the assay were: G-CSF, GM-CSF, GRO/KC/CXCL1, IFN-, IL-1α, IL-1β, IL-2, IL-4, IL-5, IL-6, IL-7, IL-10, IL-12 (p70), IL-13, IL-17A, IL-18, M-CSF, MCP-1/CCL2, MIP-1α/CCL3, MIP-3α/CCL20, RANTES/CCL5, TNF-α, and VEGF. Tissue homogenates diluted 1:4 and technical replicates were run in duplicates. Samples were run and analyzed following the manufacturer's instructions using the BioPlex 200 and HFT Systems (Bio-Rad Laboratories, Inc. Concentrations were calculated based on the provided standard curve; a 5-parametric fitted curve was calculated using the Bioplex Manager (Version 6.2, Build 175) analysis software. Analytes not included in graphs were either below the limit of detection of the assay or no significant changes were noted.

## Histology

For pathology scoring, slides with 5μm tissue sections were deparaffinized using an alcohol rehydration series and then stained following standard H&E staining procedures. For colorimetric ISH, fixed slides were deparaffinized with an alcohol rehydration series, then boiled in 10mM citric acid buffer (pH 6.0) to unmask antigen-binding epitopes. Tissue sections were permeabilized using 0.1% Triton X-100 detergent in PBS at room temperature, followed by a protease inhibitor treatment. RNAscope 2.5HD Assay Red or RNAscope 2.5HD Assay Brown detection kits were used in accordance with the manufacturer's instructions for ovary, uterus and placenta or liver sections, respectively (Advanced Cell Diagnostics, Inc (ACD)). The ISH probe, RNAscope 2.5 LS Probe- V-RVFV-ZH501-NP (ACD), targeted the nucleoprotein (NP) viral RNA. Slides were counterstained with hematoxylin and a coverslip was mounted with Permount (Fisher Chemical).

For IF imaging, cryo-sectioned slides were rehydrated with PBS containing 0.5% bovine serum albumin (BSA), then blocked with 5% donkey serum in PBS. Tissues were then probed with the following antibodies, in-house custom rabbit anti-RVFV nucleoprotein polyclonal antibody (Genscript) and anti-pan cytokeratin typeI/II anti-cytokeratin polyclonal antibody (Invitrogen; MA5-13156). Secondary antibody staining used fluorescently labeled Cy3- and Alexa488- anti-rabbit IgG. Sections were counter stained with DAPI, then a coverslip was mounted with gelvatol.

## Microscopy

H&E and colorimetric ISH slides were imaged using an Olympus CX41 microscope with the Levenhuk M300 base attachment. Immunofluorescence slides were imaged using the Nikon A1 confocal microscope provided by the University of Pittsburgh Center for Biological Imaging. Images were taken at 20x magnification. Denoising, contrasting and pseudo coloring were formed using the open-source image editor, Fiji with ImageJ [54].

## Pathology & viral antigen staining

H&E slides were blind scored by a licensed pathologist specializing in placental pathology. Scoring was based on severity of hemorrhage, inflammation, or necrosis on a scale of 0–4 (**Table 2**) [55,56]:

Colorimetric ISH slides were blind scored in-house with a scale of 0–3 using the following parameters: 0 = no staining, 1 = 1–30% staining, 2 = 30–60% staining, and 3 = >60% staining within each placenta section, decidua/myometrium, basal zone, or labyrinth zone. The average of three individuals' scores was used as the final score of each placenta section.

## Statistics

Two-way ANOVA with multiple comparisons were performed using Graphpad Prism 8.0.

**Table 2. Scoring system for histology.**

| | Hemorrhage | Inflammation | Necrosis |
|---|---|---|---|
| *0 –None* | None/normal | None/normal | None/normal |
| *1 –Trace* | Focal, occasional | ≤ 20 inflammatory cells per 40x HPF | Architectural changes, picnotic nuclei |
| *2 –Mild* | Multifocal and regional | 21–100 inflammatory cells per 40x HPF, mild edema | Focal necrosis |
| *3 –Moderate* | Regionally diffuse | 101–150 inflammatory cells per 40x HPF, moderate edema | Focal necrosis within different tissue structures |
| *4 –Severe* | 100% RBCs | ≥151 inflammatory cells per 40x HPF, extensive edema | Diffuse necrosis |

HPF = high powered field

## Supporting information

**S1 Fig. RVF in pregnant dams results in widespread viral infection of the liver, uterus, and placenta. (A)** Survival of SD rats infected at E14. Pregnant rats were infected with the indicated doses of RVFV ($1.5 \times 10^5$ pfu (n = 3), $2.6 \times 10^4$ pfu (n = 6), $1.4 \times 10^3$ pfu (n = 11), $1.8 \times 10^2$ pfu (n = 6), 75 pfu (n = 1)). The grey shaded area between 2-6dpi indicates the clinical window, when lethally infected dams were euthanized due to severe disease. **(B)** H&E and **(C)** RNA ISH staining (brown/magenta) for viral RNA within the liver, ovary, uterus, and placenta from uninfected and RVFV infected dams who met euthanasia criteria. Blue, white, and green arrow heads indicate hemorrhaging, necrosis, or leukocyte inflammation, respectively. Hematoxylin counterstain.
(TIF)

**S2 Fig. Rift Valley fever virus infects multiple placenta layers within the rat. (A)** Left panel: cross-section of an uninfected control placenta (E20) with ISH for RVFV viral RNA (magenta). Hematoxylin counterstained. The following structures were imaged for the right panels: (i) yolk sac, (ii) decidua and basal zone, (iii) maternal vasculature, and (iv) labyrinth zone. **(B)** RVFV-infected or **(C)** uninfected placentas at E20 were stained with anti-RVFV Gn antibodies (yellow), anti-cytokeratin antibodies (magenta), and DAPI followed by immunofluorescent microscopy. The following structures were imaged: decidua and basal zone (left), maternal vasculature (middle), and labyrinth zone (right). 20x images.
(TIF)

**S3 Fig. Inflammation and chemokine expression persist in post-partum uteruses from RVFV infected rats. (A)** Pathology scores identifying inflammation, necrosis and hemorrhage severity of H&E stained uterus of RVFV infected (n = 12) or uninfected dams (n = 5) that survived to post-partum (euthanized 18-22dpi). **(B)** Protein levels (pg/mL) of cytokines, chemokines, and growth factors within the uterus of infected (RVFV–Normal (n = 4–8), RVFV -Teratogenic (n = 2–4) and uninfected (n = 3) dams that survived to post-partum. $^{*}$ = p $<0.05$, $^{**}$ = p $< 0.01$. ND = none detected. An ANOVA with multiple comparisons was performed to determine statistical significance between the cohorts.
(TIF)

**S1 Table. Taqman gene expression assay kits used for semi-quantitative analyses.**
(TIF)

## Acknowledgments

We express our appreciation for advice on experimental design from Reagan Walker, rat art provided by Henry Ma, rat placental art by Hayley Nordstrom, and study coordination by Stacey Barrick. We also thank the Center for Biologic Imaging and the McGowan Center for Regenerative Medicine for histology support.

## Author Contributions

**Conceptualization:** Cynthia M. McMillen, Amy L. Hartman.

**Data curation:** Cynthia M. McMillen, Devin A. Boyles, Ryan M. Hoehl, Madeline M. Schwarz, Joseph R. Albe, Matthew J. Demers.

**Formal analysis:** Cynthia M. McMillen, Devin A. Boyles, Stefan G. Kostadinov, Amy L. Hartman.

**Funding acquisition:** Amy L. Hartman.

**Investigation:** Cynthia M. McMillen, Devin A. Boyles, Ryan M. Hoehl, Madeline M. Schwarz, Joseph R. Albe, Matthew J. Demers, Amy L. Hartman.

**Methodology:** Cynthia M. McMillen, Devin A. Boyles, Ryan M. Hoehl, Madeline M. Schwarz, Joseph R. Albe, Matthew J. Demers, Amy L. Hartman.

**Project administration:** Amy L. Hartman.

**Supervision:** Amy L. Hartman.

**Validation:** Amy L. Hartman.

**Visualization:** Cynthia M. McMillen, Devin A. Boyles, Amy L. Hartman.

**Writing – original draft:** Cynthia M. McMillen, Amy L. Hartman.

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
