## [Decision Letter · Decision Letter 0]

25 Aug 2022

Dear Dr. Hartman,

Thank you very much for submitting your manuscript "Congenital Rift Valley fever in Sprague Dawley rats is associated with diffuse infection and pathology of the placenta" for consideration at PLOS Neglected Tropical Diseases. As with all papers reviewed by the journal, your manuscript was reviewed by members of the editorial board and by several independent reviewers. The reviewers appreciated the attention to an important topic. Based on the reviews, we are likely to accept this manuscript for publication, providing that you modify the manuscript according to the review recommendations. 

Sincerely,

Adly M.M. Abd-Alla, Prof asso.

Academic Editor

Paulo Pimenta

Section Editor

Reviewer's Responses to Questions

**Key Review Criteria Required for Acceptance?**

**Methods**

-Are the objectives of the study clearly articulated with a clear testable hypothesis stated?

-Is the study design appropriate to address the stated objectives?

-Is the population clearly described and appropriate for the hypothesis being tested?

-Is the sample size sufficient to ensure adequate power to address the hypothesis being tested?

-Were correct statistical analysis used to support conclusions?

-Are there concerns about ethical or regulatory requirements being met?

Reviewer #1: Clearly stated aims and detailed methods are provided. Appropriate statistical analyses. More detail for euthanasia criteria/scoring (or a reference to previously published work that lists in more detail) would be helpful (weight loss cut-offs, etc). In particular criteria that tend to be met most often for this model in their hands.

Reviewer #2: This manuscript clearly defines the experimental approach and methods. The design is well considered and clearly establishes that RVFV can infected and cause teratogenic effects in this animal model. This reviewer has no concerns with the experimental design or methodologies utilized to analyze the resulting data.

**Results**

-Does the analysis presented match the analysis plan?

-Are the results clearly and completely presented?

-Are the figures (Tables, Images) of sufficient quality for clarity?

Reviewer #1: Results and figures are well presented. Some minor suggestions to figure legends to improve clarity are detailed in "Editorial and Data Presentation Modifications."

Reviewer #2: The analyses here and results presented are clear, relatively concise (it is a lot of information), and thorough. I have only two comments. 1) Throughout the manuscript the authors chose to use "Viral titer" when the results look striking and larger (e.g., line 153) and just "viral RNA levels" (e.g., line 159)... At all times from what this reviewer can discern the author utilize RT-qPCR for viral RNA detection and presumably quantitation. This is NOT a "viral titer". I think to keep things clear the authors should revert to some other nomenclature where this is more obvious. I am NOT suggestion to go back and do live-virus titrations of all the tissues... but please clear up the language so that it is obvious you are talking about viral RNA and not infectious virus. 2) Another puzzling observation was the lack of cellular inflammation amongst the placentas examined. The authors relatively down play that observation, but I think it might be actually quite telling regarding the underlying pathophysiological mechanisms leading to placental infection/pathology and eventual pup teratogenesis. Could you please add in a bit more on that section in the discussion.

**Conclusions**

-Are the conclusions supported by the data presented?

-Are the limitations of analysis clearly described?

-Do the authors discuss how these data can be helpful to advance our understanding of the topic under study?

-Is public health relevance addressed?

Reviewer #1: No major concerns with conclusions. Overall, authors do a very nice job of acknowledging limitations and identifying gaps for future research. One suggestion is to also note in text the different placental structures (ruminant vs. rodent/human), as there are several instances where the rat is suggested as a model for susceptible ruminants, and while it may apply in certain cases the placental structure differences should be clearly stated as a potential limitation in that particular application.

Reviewer #2: The conclusions are well written and are adequately supported by the presented data. The authors may want to be a bit more cautious with direct extrapolations from multiple species (rodents and livestock) to humans. While it is clear that RVFV is infecting and causing vertical transmission in these SD rats, the data from actual human placentas and other aborted materials is extremely limited. These studies are likely a key guide marker, but may not directly be applicable to humans. This reviewer recommends at least some discussion of that aspect of this animal model study. Additionally, Line 463: please modify the text from "sacrifices" to "euthanasia"... in experimental animal work we are euthanizing animals when necessary, not sacrificing them upon an altar on a mountaintop. This may seem trivial, but I think euthanasia is more appropriate verbiage for the scientific literature and when talking with non-scientific audiences.

**Editorial and Data Presentation Modifications?**

Reviewer #1: Minor suggestions to improve clarity and workflow:

-Line 123 - 126: Please state clearly when sham-infected controls were euthanized. In parallel with animals that met euthanasia criteria? All at end of study? If applicable, add to study design figure or legend.

-Figure 1: Add number of animals in groups specified and mention of sham-inoculated.

e.g., Sprague Dawley (SD) rats were inoculated subcutaneously (s.c.) with RVFV ZH501 at embryonic day 14 (E14; n = ?) or sham inoculated (n = ?). 

e.g., "Eight days later (E22), the rats delivered their pups and placentas (n = ?) were...."

-Please revisit all figure legends and add groups sizes where applicable

-consider using sham-inoculated consistently throughout text and figures; NI is not clear to reader if they served as both tissue and handling controls without looking in more detail to methods

-Line 142: suggest adding info on lethality in model - i.e., ...leading to fulminant hepatic disease "and fatal outcome in xx - xx% of rats"

-Line 148: should this be "euthanized from 2-6 dpi" vs. 2-5 dpi?

-Line 154: avg. 1.8 x 10^8 pfu/mL "equivalent"

-Line 218: Please consider use of term "longitudinal" evaluations, this suggests these were predetermined time points with sufficient groups sizes at each time point vs. animals that met end-point criteria at different times. These analyses more accurately look at tropism in the placenta of animals that succumb to RVFV infection (at different times). For same reason, this study design does not necessarily "identify earliest structure targeted by RVFV", but rather structures infected in animals that reach criteria at different time points (some earlier than others).

-Above comment also applies to Figure 4 legend - suggest removing "longitudinal" and just stating "Pathology in infected placentas from dam succumbing to disease from E16-E20...

-Suggest clarifying in figure headers when analysis is focused solely on end-point tissues...e.g., for figure 3: RVFV infection is detected in the placenta of dams "that succumb to infection" as early as E16 (2dpi)

-Figure 5/6: both list a range of group sizes per time point which is not helpful as the data is analyzed in aggregates (NI, normal, teratogenic) - can group sizes be given for these instead to align with analyses presented?

Reviewer #2: This reviewer has no concerns with the data or figures etc as presented.

**Summary and General Comments**

Reviewer #1: McMillen et al., present a substantial body of data from well executed studies to investigate RVFV in the pregnant rat model. These data are a welcome and much needed addition to the literature. While the manuscript is well written, there are some areas of the text that could be revisited for clarity. I believe the paper would benefit from consideration for these points and minor editing where applicable. Overall, great work and important advancement in the field!

Reviewer #2: This manuscript by McMillen et al., continues the high-quality and rigorous work from the University of Pittsburgh CVR on Rift Valley fever virus over the past several years. This study is highly significant and lays the groundwork necessary to understand the pathophysiologic (and molecular) mechanisms of RVFV infection of placental and fetal tissues in an amenable animal model. Overall the experiments were well conceived, executed, and the data analyses appear to be adequate for a firm advance of our state of knowledge regarding this emerging viral zoonoses. I have only minor comments and suggested edits. The authors are to be congratulated for a thorough and rigorous examination of experimental RVFV ZH501 infection during gestation.

PLOS authors have the option to publish the peer review history of their article (what does this mean?). If published, this will include your full peer review and any attached files.

Reviewer #1: No

Reviewer #2: No

Figure Files:

Data Requirements:

Reproducibility:

References

---

## [Editor Report · Decision Letter 1]

17 Oct 2022

Dear Dr. Hartman,

We are pleased to inform you that your manuscript 'Congenital Rift Valley fever in Sprague Dawley rats is associated with diffuse infection and pathology of the placenta' has been provisionally accepted for publication in PLOS Neglected Tropical Diseases.

Best regards,

Adly M.M. Abd-Alla, Prof asso.

Academic Editor

Paulo Pimenta

Section Editor

---

## [Editor Report · Acceptance letter]

24 Oct 2022

Dear Dr. Hartman,

We are delighted to inform you that your manuscript, "Congenital Rift Valley fever in Sprague Dawley rats is associated with diffuse infection and pathology of the placenta," has been formally accepted for publication in PLOS Neglected Tropical Diseases.

Best regards,

Shaden Kamhawi

co-Editor-in-Chief

Paul Brindley

co-Editor-in-Chief
